# Zinc utilisation, trends, and predictors among under-five children with diarrhoea in Ethiopia: A pooled analysis

**Girma Beressa** [ID]*

Department of Public Health, Madda Walabu University, Goba, Ethiopia

* gberessa@gmail.com

**Data Availability Statement:** "Third party data was obtained for this study from The DHS Program (https://dhsprogram.com/). Data may be requested from The DHS Program after creating an account and submitting a concept note. More access

## Abstract

Zinc has a significant benefit in saving children's lives. It decreases severity, diarrhoeal duration, and death rates. However, evidence on zinc utilisation, trends, and predictors among under-five children with diarrhoea in Ethiopia was sparse and inconclusive. This study aimed to assess the pooled zinc utilisation, trends, and predictors among under-five children with diarrhoea in Ethiopia. This study used Ethiopian demographic and health survey (EDHS-2005–2016) data with a total weighted sample size of 29,525 among under-five children with diarrhea. A multilevel mixed-effects logistic regression analysis was used to identify predictors of zinc utilisation. An adjusted odds ratio (AOR) along with a 95% confidence interval (CI) was used to estimate the strength of the association. The pooled zinc utilisation among under-five children in Ethiopia was 8.96% (95% CI: 7.44, 10.76%). In Ethiopia, the proportion of zinc utilisation by under-five children decreased from 0.22% (95% CI: 0.07, 0.74%) in EDHS 2005 to 0.04% (95% CI: 0.00, 0.22%) in EDHS 2011, and sharply increased to 33.60% in EDHS 2016. After adjusting for other background characteristics, having mothers complete primary education [AOR = 3.16, 95% CI: 1.57, 6.35] was a significant predictor of zinc utilisation among under-five children with diarrhea. The findings revealed that zinc utilisation was considerably low among Ethiopian under-five children with diarrhoea compared to reports from low-income countries. Ethiopia should pursue strategies to boost zinc utilisation in this group of population.

## Introduction

Zinc is a trace element required for a healthy immune system, and its lack might make a person more prone to illness and disease [1]. Zinc is not stored in the body; therefore, its amount is determined by dietary intake, absorption, and loss. That is, zinc-deficient states may arise in children with severe diarrhoea due to intestinal loss [2]. Zinc therapy for acute diarrhoea improves immunological function, intestinal structure, and epithelial healing [3]. The United Nations Children's Fund (UNICEF) and the World Health Organisation (WHO) have recommended managing children's diarrhoea with continuous feeding, oral rehydration salts, and zinc supplements since 2004 [4]. The recommended dosage of zinc for the treatment of

information can be found on The DHS Program website (https://dhsprogram.com/data/Access-Instructions.cfm).The data set is openly available upon permission from the MEASURE DHS website (https://www.dhsprogram.com/data/available-datasets.cfm). The authors confirm that interested researchers would be able to access these data in the same manner as the authors. The authors also confirm that they had no special access privileges that others would not have.

**Funding:** The author(s) received no specific funding for this work.

**Competing interests:** The author has no conflicts of interest.

diarrhoea is 20 mg per day for children over six months old and 10 mg per day for those under six months old for roughly 10–14 days [3]. Zinc has been utilised for managing diarrhoea since then, saving many children's lives. It lowers diarrhoea length by 19.7%, mortality by 23%, and the severity of initial and recurrent bouts in 2–3 months after supplementation [5, 6]. In Africa, the prevalence of excellent diarrhoea care (including oral rehydration therapy, zinc, and increased eating frequency) is low, ranging from 17% in Cote d'Ivoire to 38% in Niger [7]. Another study in Kenya found that only 18% of careers or mothers give zinc for the management of diarrhoea in their sick children [8]. In East Africa, the proportion of zinc utilisation among under-five children with diarrhoea was 21.54% [9].

In Ethiopia, 46 and 33% of under-five children with diarrhoea received oral rehydration treatment and zinc, respectively, whereas only 17% received a combination of these [10]. Previous studies revealed that mothers with formal education, wealth index, community women education, media exposure, and mothers from five and above household size were associated with zinc utilisation among under-five children with diarrhoea [9, 11]. The zinc utilisation was low among under-five children with diarrhoea in Ethiopia [12].

This study's findings can foster initiatives and policies that align with the United Nations' Sustainable Development Goals (SDGs) [13]. Identifying significant individual and community-level predictors of zinc utilisation is crucial to improving zinc utilisation in Ethiopia. Therefore, this study aimed to assess the pooled zinc utilisation, trends, and predictors among under-five children with diarrhoea in Ethiopia using Ethiopia Demographic and Health Survey (EDHS-2005-2016) data.

## Methods

### Study design, data source, and participants

The data used for this study were accessed from the Ethiopian Demographic and Health Survey (EDHS) 2005–2016 data, which was used in a community-based cross-sectional study. Ethiopia has nine regional states (Afar, Amhara, Benishangul-Gumuz, Gambella, Harari, Oromia, Somali, the Southern Nations Nationalities and People's Region (SNNPR), and Tigray), as well as two city administrations (Addis Ababa and Dire Dawa). The emerging regions of Ethiopia were Afar, Somali, Benishangul-Gumuz, and Gambella [14]. The source population was all under-five children having diarrhoea who lived with their mother, whereas the study population was all chosen or sampled living under-five children having diarrhoea who lived with their mother in the selected areas. Mothers or carers with under-five children who were sick or had hearing impairments during enumeration time were excluded from the study.

The standard EDHS data set has a large sample size, which helps to generate parameters [15]. To collect data from nine regional states and two municipal administrations, the EDHS employed a two-stage sampling technique. In the first phase, clusters of enumeration areas (EAs) (urban and rural areas) were chosen with a probability proportionate to their size using the 2005–2016 primary health care (PHC) frameworks, with independent selection in each sample stratum. Details were described elsewhere [10, 16, 17]. The second step is a systematic sampling of residences within each cluster, or EA, followed by interviews with selected mother-child pairs. The EDHS findings, published on the Measure DHS website, provide a comprehensive sample procedure (www.dhsprogram.com).

### Data collection instruments

Child sex, maternal age, marital status, religion, maternal education, wealth index, and media exposure were individual-level factors, whereas residence and regions were community-level factors [12]. Regions of the country were classified as developed and emerging based on

existing evidence [18]. Women's media exposure was assessed using three variables: frequency of reading newspapers or magazines, listening to radio, and watching TV. Women were classified as yes if they were exposed to all or any of the three, and no if they were not exposed to at least one [15]. The wealth index was computed using principal component analysis (PCA). Details are described elsewhere [12, 16, 17].

### Outcome assessment

The zinc utilisation was assessed by asking the mother if zinc was administered to her child at any point since the start of diarrhea. The response variable was dichotomised as yes (1) or no (0) [9, 19].

### Data processing and analysis

Stata$^{TM}$ 14 was used to conduct the data analysis. Descriptive statistics, such as frequency and percentage, were employed to describe informants. The proportions and frequencies were weighted. All analyses used the individual sample weight (v005/1,000,000) to account for over- and under-sampling. The EDHS dataset is hierarchical, with children nested in households and households within clusters. A correlation matrix was used to assess multicollinearity among predictors.

There were four models used: the null model (no predictors), model 1: individual-level variables, model 2: community-level factors, and model 3: individual and community-level factors. Multilevel mixed-effects logistic regression analyses were carried out to determine the relationship between individual and community-level characteristics and zinc utilisation (yes = 1, no = 0). The data were analysed using a complex sample survey multilevel mixed-effects logistic regression analysis (melogit [pweight = swt] || v001]. The Stata command "svyset" was used to establish survey data and estimate the percentage of zinc utilisation. The strength of the link between predictors and endpoint variables was evaluated using the adjusted odds ratio (AOR) along with a 95% confidence interval (CI). The model's goodness-of-fit was evaluated using the Akaike Information Criterion (AIC), the Bayesian Information Criterion (BIC), and the log-likelihood ratio (LLR). The model with the lowest AIC and BIC, as well as the highest LLR, was chosen as the best fit [20].

The Median Odds Ratio (MOR), defined as the median value of the odds ratio between the areas at the lowest and highest risk when two clusters are randomly selected, was used to quantify variation. MOR = e0.95$\sqrt{VA}$ or exp. [$\sqrt{(2 \times VA)} \times 0.6745$], where VA represents the area-level variation [21]. The median odds ratio (MOR) is a statistical metric that evaluates the relationship between an exposure and an outcome in a logistic regression model. Its interpretation is similar to that of the ordinary odds ratio, but it focusses on the odds of a randomly picked individual from one group versus a randomly selected individual from another group. MOR = 1: This result implies that the chances of the outcome are the same between the two groups being compared. In other words, the exposure doesn't change the chances of the result. MOR > 1: A MOR larger than 1 indicates that the exposure is associated with an increased likelihood of the result happening. This suggests a positive relationship, which means that as exposure grows, so does the chance of the consequence. If the MOR is less than 1, it indicates that the exposure is associated with a decreased likelihood of the outcome [22]. The proportional change in variance (PCV) measures the variance in zinc utilisation among under-five children, which is explained by several variables. The PCV is calculated as Vnull-VA/Vnull* 100. Where Vnull is the initial model's variance and VA is the model's variance with added terms. The intraclass correlation coefficient (ICC) estimates the variation in response variable between clusters. It is computed as ICC = VA ÷ VA + 3.29 * 100%, where VA = area/cluster level variance [21]. A p value less than 0.05 was declared statistically significant.

### Ethical consideration

The study did not require ethics clearance or participant consent. Ethical permission was received from Measure DHS using a data access request form. The EDHS data is available to the general public in various formats upon request from the Measure DHS website (www. measuredhs.com). All approaches adhered to the key concepts of the Helsinki Declaration [23].

## Results

### Sociodemographic and economic factors

This study employed a weighted sample of 29,525 mother-child pairs from the EDHS 2005–2016 (EDHS-2005 = 9,808, EDHS-2011 = 11,469, and EDHS-2016 = 8,248) datasets. Most of the children's mothers (EDHS-2005 = 7,603 (77.16%), (EDHS-2011 = 8130 (69.86%), and (EDHS-2005 = 6,830 (64.25%)) were uneducated. The majority (EDHS-2005 = 2,525 (25.63%), (EDHS-2011 = 3,620 (31.11%), and (EDHS-2016 = 3,989 (37.52%)) of the study subjects were in the poorest wealth quintile. Most (EDHS-2005 = 8,497 (86.24%), (EDHS-2011 = 9,655 (82.96%), and (EDHS-2016 = 8,657 (81.43%) respondents were residents in the Oromia region of Ethiopia (Table 1).

### Proportion and trends of zinc utilisation

The pooled proportion of zinc utilisation among under-five children with diarrhoea in Ethiopia was 8.96% (95% CI: 7.44, 10.76%). In Ethiopia, the proportion of zinc utilisation by under-five children decreased from 0.22% (95% CI: 0.07, 0.74%) in EDHS 2005 to 0.04% (95% CI: 0.00, 0.22%) in EDHS 2011, and sharply increased to 33.60% in EDHS 2016.

### Individual and community-level predictors of zinc utilisation among under-five children

The interclass correlation coefficient (ICC) of the null model was 30.25%. This suggested that under-five children had varying zinc utilization across clusters. The full model had the lowest AIC and highest LLR; hence, it was selected as the best-fit model. The multivariable multilevel mixed-effects logistic regression analyses revealed that having mothers complete primary education [AOR = 3.16, 95% CI: 1.57, 6.35], EDHS-2005 [AOR = 0.001, 95% CI: 0.0001, 0.004], and EDHS-2011 [AOR = 0.0001, 95% CI: 0.00002, 0.001] were significantly associated with zinc utilisation among under-five children with diarrhoea (Table 2).

## Discussion

This study aimed to assess the pooled zinc utilisation, trends, and predictors among under-five children with diarrhoea in Ethiopia. The findings clearly pointed out that the pooled proportion of zinc utilisation among under-five children with diarrhoea in Ethiopia was 8.96%. The findings revealed that under-five children having mothers who completed primary education was significantly associated with zinc utilisation among under-five children with diarrhoea.

The pooled proportion of zinc utilisation among under-five children with diarrhoea in Ethiopia was 8.96%. This finding was lower than studies conducted in Ethiopia [9, 11, 24, 25], East Africa [9], Uganda [9], Nigeria [26, 27], and SSA [28]. This could be attributed to differences in media outlet exposure and socioeconomic status.

Zinc utilisation in Ethiopia, as measured by the Ethiopian Demographic and Health Surveys (EDHS), fluctuated significantly between 2005, 2011, and 2016. Understanding these

**Table 1. Sociodemographic and economic factors of study subjects, Ethiopia, EDHS 2005–2016 (N = 29,525).**

| Variables | EDHS-2005 | EDHS-2011 | EDHS-2016 | Pooled EDHS 2005–2016 |
|---|---|---|---|---|
| | Frequency (%) | Frequency (%) | Frequency (%) | Frequency (%) |
| **Child's sex** | | | | |
| Male | 5,023 (50.98) | 3,976 (51.37) | 5,477 (51.52) | 15,174 (51.39) |
| Female | 4,830 (49.02) | 5,659 (48.63) | 5,154 (48.48) | 14,351 (48.61) |
| **M-age (years)** | | | | |
| 15–19 | 533 (5.41) | 512 (4.40) | 404 (3.80) | 1,345 (4.56) |
| 20–34 | 6,893 (69.96) | 8, 343 (71.69) | 7,686 (72.30) | 20,993 (71.10) |
| $\geq$ 35 | 2, 427 (24.63) | 2,783 (23.91) | 2,541 (23.90) | 7,187 (24.34) |
| **Marital status** | | | | |
| Married | 9,071 (92.06) | 10,177 (87.43) | 9,893 (93.06) | 26,822 (90.85) |
| Others | 782 (7.94) | 1,461 (12.55) | 738 (6.94) | 2,703 (9.15) |
| **Religion** | | | | |
| Orthodox | 3,896 (39.54) | 3,614 (31.05) | 3,078 (28.95) | 10,013 (33.91) |
| Protestant | 1,776 (18.02) | 2,237 (19.22) | 1859 (17.49) | 5,239 (17.74) |
| Muslim | 3,844 (39.01) | 5,439 (46.73) | 5,439 (51.16) | 13,596 (46.05) |
| Others # | 337 (3.42) | 348 (2.99) | 255 (2.40) | 677 (2.29) |
| **Maternal education** | | | | |
| No education | 7,603 (77.16) | 8,130(69.86) | 6,830 (64.25) | 21,415 (72.53) |
| Primary education | 1,548 (15.71) | 2,927 (25.15) | 2,677 (25.18) | 6,350 (21.51) |
| Secondary education | 631 (6.40) | 386 (3.32) | 733 (6.89) | 1,367 (4.63) |
| Higher education | 71 (0.72) | 195 (1.68) | 391 (3.68) | 393 (4.63) |
| **Wealth quintiles** | | | | |
| Poorest | 2,525 (25.63) | 3,620 (31.11) | 3,989 (37.52) | 9, 355 (31.69) |
| Poorer | 1,846 (18.74) | 2,109 (18.12) | 1,779 (16.73) | 5,427 (18.38) |
| Middle | 1,837 (18.64) | 1,870 (16.07) | 1,465 (13.78) | 4,969 (16.83) |
| Richer | 1,670 (16.95) | 1,868 (16.05) | 1,308 (12.30) | 4,964 (15.80) |
| Richest | 1,975 (20.04) | 2,171 (18.65) | 2,090 (19.66) | 5110 (17.31) |
| **Residence** | | | | |
| Urban | 1,356 (13.76) | 1,983 (17.03) | 1,974 (18.54) | 4,194 (14.20) |
| Rural | 8,497 (86.24) | 9,655(82.96) | 8,657 (81.43) | 25,331 (85.80) |
| **Region** | | | | |
| Developed | 7,409 (75.20) | 7,618 (65.46) | 6,477 (60.90) | 19, 705 (66.74) |
| Developing | 2,444 (24.81) | 4,020 (34.54) | 4,157 (39.10) | 9, 820 (33.26) |

M-age: maternal age # Others: catholic, traditional, and others; EDHS: Ethiopia demographic and health survey

patterns entails examining numerous educational initiatives, socioeconomic and health-related variables, health policy changes, and community involvement strategies that developed throughout time and affected zinc supplementation practices.

The EDHS 2005 data corresponded with increased knowledge of the advantages of zinc supplementation for diarrhoea treatment, owing primarily to recommendations from organisations such as UNICEF and WHO. These guidelines emphasised zinc's function in lowering diarrhoea length and fatality rates, which may have contributed to increasing use during this period [19]. In the early 2000s, there was a big push for community-based health education, which most likely boosted carers' understanding of zinc supplementation. This instructional emphasis may have diminished by 2011, resulting in a decrease in utilisation rates. Changes in home dynamics, such as higher household sizes or economic issues, may have influenced

**Table 2. Multivariable multilevel mixed-effects logistic regression analyses of zinc utilisation and predictors among under-five children with diarrhoea, Ethiopia, 2005–2016 (N = 29,525).**

| Variable | Null model[a] | Model 1[b] | Model 2[c] | Model 3[d] |
|---|---|---|---|---|
| | AOR(95% CI) | AOR (95% CI) | AOR (95% CI) | AOR (95% CI) |
| **Child's sex** | | | | |
| Male | | Ref | | Ref |
| Female | | 1.41 (0.91, 2.18) | | 1.30 (0.77, 2.20) |
| **M-age (months)** | | | | |
| 15–19 | | Ref | | Ref |
| 20–34 | | 1.29 (0.44, 3.78) | | 0.67 (0.19, 2.44) |
| ≥ 35 | | 1.87 (0.59, 5.95) | | 1.45 (0.34, 6.12) |
| **Marital status** | | | | |
| Married | | 1.80 (0.75, 4.33) | | 1.32 (0.38, 4.55) |
| Others | | Ref | | Ref |
| **Religion** | | | | |
| Orthodox | | Ref | | Ref |
| Protestant | | 0.76 (0.39, 1.48) | | 0.59 (0.27, 1.24) |
| Muslim | | 1.73 (0.88, 3.41) | | 1.27 (0.55, 2.92) |
| Others # | | 0.21 (0.02, 1.93) | | 0.10 (0.006, 1.97) |
| **Maternal education** | | | | |
| No education | | Ref | | Ref |
| Primary education | | 4.89 (2.91, 8.21) | | **3.16 (1.57, 6.35)***  |
| Secondary education | | 4.34(1.52, 12.32) | | 1.62 (0.47, 5.64) |
| Higher education | | 2.72 (0.35, 20.86) | | 0.48 (0.05, 4.13) |
| **Wealth quintile** | | | | |
| Poorest | | Ref | | Ref |
| Poorer | | 0.93 (0.50, 1.70) | | 0.80 (0.36, 1.77) |
| Middle | | 0.82 (0.42, 1.59) | | 0.60 (0.25, 1.46) |
| Richer | | 0.79 (0.41, 1.50) | | 0.71 (0.29, 1.70) |
| Richest | | 1.13 (0.50, 2.75) | | 1.26 (0.35, 4.49) |
| **Media exposure** | | | | |
| No | | Ref | | Ref |
| Yes | | 0.72 (0.45, 1.17) | | 1.66 (0.86, 3.20) |
| **Residence** | | | | |
| Urban | | | Ref | Ref |
| Rural | | | 0.27 (0.12, 0.60) | 0.47 (0.14, 1.52) |
| **Region** | | | | |
| Developed region | | | 0.72 (1.39, 1.32) | 0.75 (0.31, 1.81) |
| Developing region | | | Ref | Ref |
| **Survey time** | | | | |
| EDHS 2005 | | | 0.002 (0.001,0.004) | **0.001(0.0001,0.004)***  |
| EDHS 2011 | | | 0.001(0.00, 0.002) | **0.0001(0.00002, 0.001)****  |
| EDHS 2016 | | | Ref | Ref |
| **Random effects** | | | | |
| Variance (SE) | 5.28 (0.84) | 5.64 (1.01) | 3.14 (0.75) | 3.41 (0.94) |
| ICC% | 30.25 | 30.57 | 17.34 | 16.26 |
| PCV% | Ref | -6.56 | 40.50 | 35.51 |
| MOR | 8.84 | 9.49 | 5.37 | 5.75 |
| **Model fit statistics** | | | | |

*(Continued)*

**Table 2.** (Continued)

| Variable | Null model[a] | Model 1[b] | Model 2[c] | Model 3[d] |
|---|---|---|---|---|
| | AOR(95% CI) | AOR (95% CI) | AOR (95% CI) | AOR (95% CI) |
| AIC | 2194.76 | 2106.35 | 1383.64 | 1345.94 |
| BIC | 2207.32 | 2213.15 | 1421.33 | 1477.87 |
| Log likelihood | -1163.87 | -1107.92 | -700.11 | -669.65 |

* Statistically significant p value < 0.05

** p value < 0.001; Ref: Reference; AOR: adjusted odds ratio; CI: confidence interval; SE: standard error; ICC: intraclass correlation, PCV: proportional change in variance; MOR: median odds ratio; [a]model without predictors; [b]adjusted for individual-level factors; [c]adjusted for community-level factors; [d]full model; EDHS: Ethiopia demographic and health survey; # Others: catholic, traditional, and others

access to healthcare services, particularly zinc supplements, resulting in lower utilisation rates in 2011 than in 2005 [19].

By 2016, there had been renewed health campaigns urging the use of zinc in conjunction with oral rehydration solutions to treat diarrhoea. The Ethiopian government and health organisations have increased efforts to incorporate zinc into routine treatment procedures for paediatric diarrhoea [11, 29]. The 2016 survey found a link between maternal education and zinc use. Mothers with formal education were substantially more likely to use zinc supplements. As educational programs improved over time, more women learnt about the need of zinc, which led to greater use [12, 19]. Increased media exposure (radio, television, and newspapers) also helped to raise knowledge about the advantages of zinc. The prevalence of efficient health communication methods probably led to the substantial increase recorded in 2016 [19]. Increased community health services and access to healthcare facilities increased the availability of zinc supplements, permitting better use rates among families with young children suffering from diarrhoea [11, 29].

Children born to mothers who completed a primary education level were 3.16 times more likely to utilize zinc compared to children born to mothers who did not have an education. This study supported researches carried out in Ethiopia [11, 19, 30, 31], East Africa [9], Nigeria [32], SSA [28]. This might be because educated women are exposed to a variety of media channels and realize the importance of zinc for health. Moreover, educated women provide mothers with the chance to seek medical advice and provide prescribed medicines, including zinc, for their ill children.

## Policy implications

This study's findings have significant implications for clinical and public health intervention. Despite the benefits of zinc, limited zinc utilization indicates a policy gap in Ethiopia's maternal and child nutritional status. The findings have implications for improving food and nutrition policy, national nutrition strategy, and national nutrition programme by increasing mothers' educational levels in order to boost zinc utilization among under-five children [33–35].

To improve zinc use in under-five children with diarrhoea, numerous policy strategies might be implemented: Governments and health organisations should engage in mass media efforts to educate parents on the benefits of zinc supplementation in addition to oral rehydration salts (ORS). Such campaigns can target less educated communities to enhance understanding and habits related to child health [19, 32]. Establishing mechanisms to track zinc supplementation rates can aid in identifying gaps in service delivery and informing future initiatives.

### Strengths and limitations of the study

This study was based on nationally representative large weighted data and used appropriate statistical analysis techniques. However, the data were collected via self-reports, which may have contributed to recall bias. Moreover, because the study is cross-sectional, a cause-and-effect link may not be established. Furthermore, the EDHS survey does not include some data on dietary intake of zinc (dietary sources of zinc), which affects zinc utilisation. Women do not always offer zinc tablets because they believe that zinc-rich food is a substitute for zinc tablets.

### Conclusion

The findings revealed that the pooled zinc utilisation among under-five children with diarrhoea in Ethiopia was low compared to reports from low-income countries. The findings indicated that having mothers complete primary education was significantly associated with zinc utilisation among under-five children with diarrhea. Ethiopia should pursue strategies to boost zinc utilisation throughout these important phases of growth and development. The conclusions of this study would have ramifications for Ethiopian decision-makers to improve zinc utilisation. Further research, such as a randomized controlled trial (RCT), is warranted to attain robust findings.

### Acknowledgments

The author thanks the Demographic Health Survey Data Archivist for sharing the dataset with me.

### Author Contributions

**Conceptualization:** Girma Beressa.

**Formal analysis:** Girma Beressa.

**Methodology:** Girma Beressa.

**Software:** Girma Beressa.

**Validation:** Girma Beressa.

**Writing – original draft:** Girma Beressa.

**Writing – review & editing:** Girma Beressa.

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
