## [Decision Letter · Decision Letter 0]

7 Oct 2024

PONE-D-24-39627Zinc utilisation, trends, and predictors among under-five children with diarrhoea in Ethiopia: a pooled data analysisPLOS ONE

Dear Dr. Beressa,

Thank you for submitting your manuscript to PLOS ONE. After careful consideration, we feel that it has merit but does not fully meet PLOS ONE’s publication criteria as it currently stands. Therefore, we invite you to submit a revised version of the manuscript that addresses the points raised during the review process.

Please submit your revised manuscript by Nov 21 2024 11:59PM. If you will need more time than this to complete your revisions, please reply to this message or contact the journal office at plosone@plos.org. Please include the following items when submitting your revised manuscript:A rebuttal letter that responds to each point raised by the academic editor and reviewer(s). You should upload this letter as a separate file labeled 'Response to Reviewers'.A marked-up copy of your manuscript that highlights changes made to the original version. You should upload this as a separate file labeled 'Revised Manuscript with Track Changes'.An unmarked version of your revised paper without tracked changes. You should upload this as a separate file labeled 'Manuscript'.

We look forward to receiving your revised manuscript.

Kind regards,

Yitagesu Habtu Aweke, Ph.D

Academic Editor

PLOS ONE

Journal Requirements: When submitting your revision, we need you to address these additional requirements. 1. Please ensure that your manuscript meets PLOS ONE's style requirements, including those for file naming. The PLOS ONE style templates can be found at https://journals.plos.org/plosone/s/file?id=wjVg/PLOSOne_formatting_sample_main_body.pdf and https://journals.plos.org/plosone/s/file?id=ba62/PLOSOne_formatting_sample_title_authors_affiliations.pdf 2. We noticed you have some minor occurrence of overlapping text with the following previous publication(s), which needs to be addressed: https://bmcpublichealth.biomedcentral.com/articles/10.1186/s12889-020-09541-4 In your revision ensure you cite all your sources (including your own works), and quote or rephrase any duplicated text outside the methods section. Further consideration is dependent on these concerns being addressed.

Reviewers' comments:

Reviewer's Responses to Questions

**Comments to the Author**

1. Is the manuscript technically sound, and do the data support the conclusions?

Reviewer #1: Partly

Reviewer #2: Yes

2. Has the statistical analysis been performed appropriately and rigorously? 

Reviewer #1: Yes

Reviewer #2: Yes

3. Have the authors made all data underlying the findings in their manuscript fully available?

Reviewer #1: No

Reviewer #2: No

4. Is the manuscript presented in an intelligible fashion and written in standard English?

Reviewer #1: Yes

Reviewer #2: No

5. Review Comments to the Author

Reviewer #1: The paper is clearly structured and the topic of this research is very interesting to

address the public health problem of diarrhea and to show policy gaps on zinc

utilization.

1. In the introduction, lines 51-55 describe the utilization of zinc and its predictors. What are your research gaps? Clearly state the research gap in the introduction section

after line 55.

2. Check line 75; statement of size using the 2005-2019 primary health care data. However, you were using EDHS data from 2005-2016.

3. Write some statement about exclusion criteria?

4. Would you consistently use the term "under five" or "under-five" throughout your

document?

5. How do you select study participants from urban and rural residences to avoid

geographical effects?

6. What is your study population: under-five children with mothers, or under-five

children with diarrhea and their mothers? Please write clearly.

7. Write briefly on the MOR value like =1 or ≥1. indicates?

8. On lines 152-153, check whether the AOR is within the 95% CI, and also verify

this against the table.

9.On limitation, line 189, you stated that there is limited literature. However, other

references are available in Africa, including Burundi, Comoros, Kenya, Ethiopia, Madagascar, Malawi, Tanzania, Uganda, and Zimbabwe. References

1. Seifu BL, Legesse BT, Yehuala TZ, Kase BF, Asmare ZA, Mulaw GF, Tebeje

TM, Mare KU. Factors associated with the co-utilization of oral rehydration

solution and zinc for treating diarrhea among under-five children in 35 sub- saharan Africa countries: a generalized linear mixed effect modeling with robust

error variance. BMC Public Health. 2024 May 16;24(1):1329. 2. Kassa SF, Alemu TG, Techane MA, Wubneh CA, Assimamaw NT, Mulualem G, et al. The co-utilization of oral rehydration solution and zinc for treating Diarrhea

and its Associated factors among under-five children in Ethiopia: further analysis

of EDHS 2016 the co-utilization of oral. Rehydration Solution and Zinc for

Treating Diarrhea; 2022.

3. Yimenu DK, Kasahun AE, Chane M, Getachew Y, Manaye B, Kifle ZD. Assessment of knowledge, attitude, and practice of child caregivers towards oral

rehydration salt and zinc for the treatment of diarrhea in under 5 children in

Gondar town. Clin Epidemiol Glob Heal [Internet]. 2022;14:100998. 4. Ayele E, Tasew H, Mariye T, Teklay G, Alemayhu T, Mesfin F. Zinc Utilization

and Associated Factors Among Under-five Children Having Acute Diarrhea in

Kebri-dehar Town, Somali Region, Ethiopia2017. Medicine. 2020; 4(1):15–9. 5. Ugwu J, Ezeagu I, Ibegbu M. Awareness and practice of zinc therapy in diarrheal

management amongunder-five caregivers in Enugu State, Nigeria. International

Journal of Medicine and Health Development. 2019; 24(2):63–9

6. Ajayi D.T BO, Ijaola T.T. Oke O.A. and Fabiyi G.A. Determinants of Oral

Rehydration Solution and Zinc Use Among Under-Five Children for The

Management of Diarrhea in Abeokuta, Nigeria. Archives of Basic and Applied

Medicine 2019; 7:35–9. 7. Emmanuel Firima BF. Knowledge and recommendation of oral rehydration

solution and zinc for management of childhood diarrhoea among patent and

proprietary medicine vendors in Port Harcourt, Nigeria. Journal of Global Health

Reports. 2020; 4:1–12

10.Add other limitation on the nature of the EDHS survey does not include data on

dietary intake of zinc (dietary sources of zinc) which affect the utilization. Sometimes

women do not give zinc tablets because they think that food rich in zinc is a substitute

of zinc tablets.

11.Discuss the trends in zinc utilization from 2005 to 2016 and explain why zinc

utilization was higher in 2005 compared to 2011 and why there was a sharp increase

in zinc utilization in the 2016 EDHS data.

12.Conclusion: The prevalence of zinc utilization is low. Compared to what? Write

briefly by comparing with the global or national recommendations.

13. Discussion were shallow and revised this section by adding updated references.

14.Your justification on line 170 regarding the difference between socioeconomic

status and cultural status was not convincing. Both studies use EDHS data and are

from the same country? How do you see?

Reviewer #2: • The manuscript does not meet the standards of written English. We suggest having the paper edited by language professionals or native speakers.

• In Table 1, the author classified regions as developed and undeveloped. What criteria did the author use to categorize the country's regions into these groups? The author needs to justify the criteria.

• The author must construct tables appropriately; for example, table 2 is not constructed correctly.

• The authors have to ensure that the methodology section clearly explains the response and covariates of the study.

• The author needs to provide a more detailed discussion of their findings and compare and contrast them with the existing literature in the study's discussion section.

• Major concern: The author considers DHS data from 2005-2016, but it's worth noting if they observe the trend of zinc utilization, which I didn't see in the paper.

• If the authors addressed the issues mentioned above, I would recommend accepting and publishing the manuscript after revision.

6. PLOS authors have the option to publish the peer review history of their article (what does this mean?). If published, this will include your full peer review and any attached files.

Reviewer #1: No

Reviewer #2: **Yes: **Daniel Biftu Bekalo

---

## [Author Response · Author response to Decision Letter 0]

16 Oct 2024

Zinc utilisation, trends, and predictors among under-five children with diarrhoea in Ethiopia: a pooled data analysis: PONE-D-24-39627

The author would like to acknowledge the Editor and reviewers for giving me an opportunity to revise the manuscript. 

 Response to editor: The author made changes to the manuscript.

Reviewers' comments:

Reviewer #1: The paper is clearly structured and the topic of this research is very interesting to

address the public health problem of diarrhea and to show policy gaps on zinc utilization.

1. In the introduction, lines 51-55 describe the utilization of zinc and its predictors. What are your research gaps? Clearly state the research gap in the introduction section

after line 55.

Authors’ response: The author made changes to the introduction section.

2. Check line 75; statement of size using the 2005-2019 primary health care data. However, you were using EDHS data from 2005-2016.

Authors’ response: The author made changes to the methods section.

3. Write some statement about exclusion criteria?

Authors’ response: The author made changes to the methods section.

4. Would you consistently use the term "under five" or "under-five" throughout your

document?

Authors’ response: The author made changes to the manuscript.

5. How do you select study participants from urban and rural residences to avoid

geographical effects?

Authors’ response: The author made changes to the manuscript.

6. What is your study population: under-five children with mothers, or under-five

children with diarrhea and their mothers? Please write clearly.

Authors’ response: The author made changes to the manuscript.

7. Write briefly on the MOR value like =1 or ≥1. indicates?

Authors’ response: The author made changes to the methods section.

8. On lines 152-153, check whether the AOR is within the 95% CI, and also verify

this against the table.

Authors’ response: The author made changes to the results section.

9. On limitation, line 189, you stated that there is limited literature. However, other

references are available in Africa, including Burundi, Comoros, Kenya, Ethiopia, Madagascar, Malawi, Tanzania, Uganda, and Zimbabwe. References

1. Seifu BL, Legesse BT, Yehuala TZ, Kase BF, Asmare ZA, Mulaw GF, Tebeje

TM, Mare KU. Factors associated with the co-utilization of oral rehydration

solution and zinc for treating diarrhea among under-five children in 35 sub- saharan Africa countries: a generalized linear mixed effect modeling with robust error variance. BMC Public Health. 2024 May 16;24(1):1329. 2. Kassa SF, Alemu TG, Techane MA, Wubneh CA, Assimamaw NT, Mulualem G, et al. The co-utilization of oral rehydration solution and zinc for treating Diarrhea and its Associated factors among under-five children in Ethiopia: further analysis of EDHS 2016 the co-utilization of oral. Rehydration Solution and Zinc for Treating Diarrhea; 2022.

3. Yimenu DK, Kasahun AE, Chane M, Getachew Y, Manaye B, Kifle ZD. Assessment of knowledge, attitude, and practice of child caregivers towards oral rehydration salt and zinc for the treatment of diarrhea in under 5 children in Gondar town. Clin Epidemiol Glob Heal [Internet]. 2022;14:100998. 4. Ayele E, Tasew H, Mariye T, Teklay G, Alemayhu T, Mesfin F. 2017. Medicine. 2020; 4(1):15–9. 5. Ugwu J, Ezeagu I, Ibegbu M. Awareness and practice of zinc therapy in diarrheal

management among under-five caregivers in Enugu State, Nigeria. International

Journal of Medicine and Health Development. 2019; 24(2):63–9

6. Ajayi D.T BO, Ijaola T.T. Oke O.A. and Fabiyi G.A. Determinants of Oral

Rehydration Solution and Zinc Use Among Under-Five Children for The

Management of Diarrhea in Abeokuta, Nigeria. Archives of Basic and Applied

Medicine 2019; 7:35–9. 7. Emmanuel Firima BF. Knowledge and recommendation of oral rehydration solution and zinc for management of childhood diarrhoea among patent and

proprietary medicine vendors in Port Harcourt, Nigeria. Journal of Global Health

Reports. 2020; 4:1–12

Authors’ response: The author made changes to the discussion section as requested.

10. Add other limitation on the nature of the EDHS survey does not include data on

dietary intake of zinc (dietary sources of zinc) which affect the utilization. Sometimes

women do not give zinc tablets because they think that food rich in zinc is a substitute

of zinc tablets.

Authors’ response: The author made changes to the discussion section.

11. Discuss the trends in zinc utilization from 2005 to 2016 and explain why zinc

utilization was higher in 2005 compared to 2011 and why there was a sharp increase

in zinc utilization in the 2016 EDHS data.

Authors’ response: The author made changes to the discussion section.

12. Conclusion: The prevalence of zinc utilization is low. Compared to what? Write

briefly by comparing with the global or national recommendations.

Authors’ response: The author made changes to the conclusion section.

13. Discussion were shallow and revised this section by adding updated references.

Authors’ response: The author has updated the discussion section.

14. Your justification on line 170 regarding the difference between socioeconomic

status and cultural status was not convincing. Both studies use EDHS data and are

from the same country? How do you see?

Authors’ response: The author expunged the cultural status for clarity purposes.

Reviewer #2: • The manuscript does not meet the standards of written English. We suggest having the paper edited by language professionals or native speakers.

Authors’ response: The entire manuscript was thoroughly edited.

• In Table 1, the author classified regions as developed and undeveloped. What criteria did the author use to categorize the country's regions into these groups? The author needs to justify the criteria.

Authors’ response: Ethiopia's classification into developed and emerging regions is primarily based on its administrative divisions and economic performance. 

References: 1.Tamirat KS, Kebede FB, Gonete TZ, Tessema GA, Tessema ZT. Geographical variations and determinants of iron and folic acid supplementation during pregnancy in Ethiopia: analysis of 2019 mini demographic and health survey. BMC Pregnancy Childbirth. 2022;22(1):127. Epub 2022/02/17. doi: 10.1186/s12884-022-04461-0. 

2. Gebremedhin T, Geberu DM, Atnafu A. Less than one-fifth of the mothers practised exclusive breastfeeding in the emerging regions of Ethiopia: a multilevel analysis of the 2016 Ethiopian demographic and health survey. BMC Public Health. 2021;21(1):18. Epub 2021/01/06. doi: 10.1186/s12889-020-10071-2. 

3. Governent of Ethiopia, Ministry of Federal affairs, emerging regions deveopment programmes: http://efaidnbmnnnibpcajpcglclefindmkaj/https://info.undp.org/docs/pdc/Documents/ETH/00047309_Emerging%20Regions%20Development%20Programme%20(00056833).pdf. 4. https://www.worldbank.org/en/country/ethiopia/overview, 

• The author must construct tables appropriately; for example, table 2 is not constructed correctly.

Authors’ response: The author amended tables.

• The authors have to ensure that the methodology section clearly explains the response and covariates of the study.

Authors’ response: The author made changes to the methods section. The study variables were operationalised based on the existing evidence.

• The author needs to provide a more detailed discussion of their findings and compare and contrast them with the existing literature in the study's discussion section.

Authors’ response: The authors made changes to the discussion section.

• Major concern: The author considers DHS data from 2005-2016, but it's worth noting if they observe the trend of zinc utilization, which I didn't see in the paper.

Authors’ response: The authors made changes to the discussion section.

---

## [Decision Letter · Decision Letter 1]

31 Oct 2024

PONE-D-24-39627R1Zinc utilisation, trends, and predictors among under-five children with diarrhoea in Ethiopia: a pooled data analysisPLOS ONE

Dear Dr. Beressa,

Thank you for submitting your manuscript to PLOS ONE. After careful consideration, we feel that it has merit but does not fully meet PLOS ONE’s publication criteria as it currently stands. Therefore, we invite you to submit a revised version of the manuscript that addresses the points raised during the review process. Please address all issues raised by reviewer 1.

Please submit your revised manuscript by Dec 15 2024 11:59PM. If you will need more time than this to complete your revisions, please reply to this message or contact the journal office at plosone@plos.org. Please include the following items when submitting your revised manuscript:A rebuttal letter that responds to each point raised by the academic editor and reviewer(s). You should upload this letter as a separate file labeled 'Response to Reviewers'.A marked-up copy of your manuscript that highlights changes made to the original version. You should upload this as a separate file labeled 'Revised Manuscript with Track Changes'.An unmarked version of your revised paper without tracked changes. You should upload this as a separate file labeled 'Manuscript'.If applicable, we recommend that you deposit your laboratory protocols in protocols.io to enhance the reproducibility of your results. Protocols.io assigns your protocol its own identifier (DOI) so that it can be cited independently in the future. For instructions see: https://journals.plos.org/plosone/s/submission-guidelines#loc-laboratory-protocols. Additionally, PLOS ONE offers an option for publishing peer-reviewed Lab Protocol articles, which describe protocols hosted on protocols.io. Read more information on sharing protocols at https://plos.org/protocols?utm_medium=editorial-email&utm_source=authorletters&utm_campaign=protocols.

We look forward to receiving your revised manuscript.

Kind regards,

Yitagesu Habtu Aweke, Ph.D

Academic Editor

PLOS ONE

Journal Requirements:

Reviewers' comments:

Reviewer's Responses to Questions

**Comments to the Author**

1. If the authors have adequately addressed your comments raised in a previous round of review and you feel that this manuscript is now acceptable for publication, you may indicate that here to bypass the “Comments to the Author” section, enter your conflict of interest statement in the “Confidential to Editor” section, and submit your "Accept" recommendation.

Reviewer #1: All comments have been addressed

Reviewer #2: All comments have been addressed

2. Is the manuscript technically sound, and do the data support the conclusions?

Reviewer #1: Yes

Reviewer #2: Yes

3. Has the statistical analysis been performed appropriately and rigorously? 

Reviewer #1: Yes

Reviewer #2: Yes

4. Have the authors made all data underlying the findings in their manuscript fully available?

Reviewer #1: Yes

Reviewer #2: No

5. Is the manuscript presented in an intelligible fashion and written in standard English?

Reviewer #1: Yes

Reviewer #2: Yes

6. Review Comments to the Author

Reviewer #1: The authors should be attentive while editing the tables. In line 169, AOR was not found within the 95% CI; please also check Table 2. Revise the discussion sections in lines 233-238 and 241-242 based on your findings. You are discussing training and cost-related activities, though your findings were not revealed on training and cost.

Reviewer #2: (No Response)

7. PLOS authors have the option to publish the peer review history of their article (what does this mean?). If published, this will include your full peer review and any attached files.

Reviewer #1: No

Reviewer #2: **Yes: **Daniel Biftu Bekalo

---

## [Author Response · Author response to Decision Letter 1]

1 Nov 2024

Zinc utilisation, trends, and predictors among under-five children with diarrhoea in Ethiopia: a pooled analysis: PONE-D-24-39627R1

The author thanks to the Editor and reviewers for giving me an opportunity to revise the manuscript. 

 Response to editor: The author revised the manuscript.

Reviewers' comments:

Reviewer #1: The authors should be attentive while editing the tables. In line 169, AOR was not found within the 95% CI; please also check Table 2. 

Author’s response: The author made changes to the results section.

Revise the discussion sections in lines 233-238 and 241-242 based on your findings. You are discussing training and cost-related activities, though your findings were not revealed on training and cost.

Author’s response: The author expunged it for clarity purposes.

---

## [Editor Report · Decision Letter 2]

6 Nov 2024

Zinc utilisation, trends, and predictors among under-five children with diarrhoea in Ethiopia: a pooled analysis

PONE-D-24-39627R2

Dear Dr. Girma,

We’re pleased to inform you that your manuscript has been judged scientifically suitable for publication and will be formally accepted for publication once it meets all outstanding technical requirements.

Kind regards,

Yitagesu Habtu Aweke, Ph.D

Academic Editor

PLOS ONE

 Your paper may benefit if you optionally address the following before publication:

Could you please revise the misspelling “ ... emerging regions of Ethiopia...” ?I don’t think the term “model building” adds anything on the title “Data processing, model building, and analysis”, “Data processing, and analysis” is more than descriptive of the issue.

---

## [Editor Report · Acceptance letter]

7 Nov 2024

PONE-D-24-39627R2 

PLOS ONE

Dear Dr. Beressa, 

I'm pleased to inform you that your manuscript has been deemed suitable for publication in PLOS ONE. Congratulations! Your manuscript is now being handed over to our production team.

Kind regards, 

on behalf of

PhD Candidate Yitagesu Habtu Aweke 

Academic Editor

PLOS ONE